

# Predicting the prognosis of hepatocellular carcinoma based on genes related to polyamine metabolism

Chengli Liu[1,2,*], Meng Pu[1,*], Yingbo Ma[1], Shuhan Zhang[1], Qike Huang[1], Haoming Li[3], Tian Xia[4] and Jingchen Zhang[4]

[1] Department of Hepatobiliary Surgery, Air Force Medical University, Air Force Medical Center, Beijing, China
[2] Air Force Clinical College, Anhui Medical University, Beijing, China
[3] Department of Hepatobiliary Surgery, Air Force Medical Center, Postgraduate Training Base of Air Force Medical Center, China Medical University, Beijing, China
[4] Department of Hepatobiliary Surgery, Air Force Medical Center, Graduate School of Hebei North University, Zhangjiakou, China
* These authors contributed equally to this work.

## ABSTRACT

**Background:** Hepatocellular carcinoma (HCC) is a highly prevalent malignant tumor worldwide. Evidence showed that polyamine metabolism plays a crucial part in the regulation of cancer onset and development, however, its clinical significance in HCC remains unclear.

**Methods:** Bulk RNA sequencing (RNA-seq) and single-cell RNA sequencing (scRNA-seq) data of HCC were collected from public databases. Polyamine metabolism-related genes (PMRGs) were obtained from the MSigDB database. The molecular subtypes of HCC were classified by ConsensusClusterPlus package, and differentially expressed genes (DEGs) of the molecular subtypes were identified by the limma package, followed by enrichment analysis with clusterProfiler package. Univariate Cox and Lasso Cox regression analyses were performed to screen core genes, construct risk model, and develop a nomogram integrating clinical characteristics for survival prediction. The obtained biomarkers were validated using *in vitro* experiments (CCK8, wound healing, and Transwell assay). The Tumor Immune Estimation Resource (TIMER), MCP-counter, and Cell Identification by Estimating Relative Subsets of RNA Transcripts (CIBERSORT) methods were employed for immune cell infiltration analysis. Finally, drug sensitivity of the HCC samples was analyzed with the oncoPredict package.

**Results:** This study identified two molecular subtypes (C1 and C2), with C2 demonstrating a more favorable prognosis. Glucose-6-phosphate dehydrogenase (*G6PD*), alcohol dehydrogenase 4 (*ADH4*), S100 calcium binding protein A9 (*S100A9*), aldo-keto reductase family 1 member B15 (*AKR1B15*) were predicted as the biomarkers for HCC. Cell experiment results showed that the expressions of *G6PD*, *AKR1B15*, and *S100A9* were all notably elevated in HuH-7 cells. Moreover, the loss of *G6PD* gene expression reduced the viability, migratory, and invasive capabilities of HCC cells. Patients with a high RiskScore had a lower survival rate than those with a low RiskScore. Scores of immune cells such as Tregs and M0 macrophages were higher in the high-risk group, and 13 drugs were found to be significantly linked to the RiskScore. Single-cell analysis showed that *G6PD* and

Corresponding author
Chengli Liu, liuchengli_kj@163.com

*S100A9* were high-expressed mainly in hematopoietic progenitor cells (HPCs) and macrophages.

**Conclusion:** In conclusion, this study screened four key genes based on PMRGs and constructed a risk model to effectively predict the prognosis of HCC, providing novel potential targets and theoretical basis for the molecular subtyping and individualized treatment of HCC.

## INTRODUCTION

Liver cancer is a frequent malignancy (*Tang et al., 2024*; *Zhou et al., 2024*) that has become a main cause of global cancer death (*Bray et al., 2024*). Hepatocellular carcinoma (HCC) is the most common primary liver cancer type that accounts for 80% to 90% of the total number of lung cancer cases, and its incidence varies significantly in different geographical regions (*Ringelhan et al., 2018*). In recent years, the treatment of HCC has been advanced through targeted therapies (*e.g.*, sorafenib and lenvatinib) to target tumor growth pathways and immune checkpoint inhibitors (*e.g.*, atilizumab and bevacizumab). Compared to traditional first-line sorafenib-based chemotherapy, these novel therapies can better improve patients' prognosis (*Greten et al., 2019*; *Cooper & Giancotti, 2019*). However, despite these advancements, the emergence of resistance to targeted therapies including sorafenib and their limited efficacy in late-stage HCC highlight an urgent need for more specific biomarkers to improve the early diagnosis of HCC (*Seyfinejad & Jouyban, 2022*; *Lin et al., 2024*).

Recent studies have reported a crucial role of polyamine metabolism in promoting tumor cell growth, proliferation, and survival. Cancer-associated dysregulation of this pathway is typically characterized by elevated polyamine levels (*Ma et al., 2021*; *Puleston et al., 2021*). Study reported that polyamine biosynthesis pathway activates hypusinated eukaryotic translation initiation factor 5A (eIF5AH), which enhances the translation of mitochondrial proteins involved in oxidative phosphorylation (OXPHOS) and tricarboxylic acid (TCA) cycle (*Puleston et al., 2019*). This metabolic shift further promotes M2 macrophage polarization, contributing to the development of an immunosuppressive tumor microenvironment (*Murray-Stewart, Woster & Casero, 2016*). Notably, HCC frequently exhibits aberrant polyamine metabolism characterized by elevated polyamine levels. This metabolic alteration not only derives tumor cell proliferation and survival, but also supports immune evasion, thereby accelerating HCC progression (*Casero, Murray Stewart & Pegg, 2018*; *Li et al., 2019*). These discoveries showed that polyamine metabolism could act as a potential indicator of tumor aggressiveness and a target for therapeutic intervention in HCC (*Liu et al., 2024*).

The present work investigated the role of polyamine metabolism-related genes (PMRGs) in the development of HCC and constructed a risk model for clinical assessment. Considering the key regulatory role of polyamine metabolism in tumor cell proliferation,

immune escape and microenvironment remodeling, this study also has important theoretical and practical significance. Based on public databases, we first screened PMRGs significantly associated with prognosis and identified prognostic features and functional pathways specific to different HCC molecular subtypes. Subsequently, key genes were screened by univariate Cox and Lasso regression analyses, and a prognostic risk model was constructed and validated. To enhance the clinical utility, a nomogram was developed integrating the risk model and clinical characteristics. The relationship between the model and immune infiltration and drug sensitivity was further analyzed, and the cellular distribution characteristics of the key genes were explored based on the single-cell data. Finally, the functions of the key genes were verified by *in vitro* experiments. This study provides new ideas and potential targets for molecular typing, precision therapy and immune intervention in HCC.

## MATERIALS AND METHODS

### Data collection and acquisition

In this study, the bulk dataset (training set) containing the RNA-seq data of The Cancer Genome Atlas (TCGA)-HCC was downloaded through TCGA Genomic Data Commons (GDC) Application Programming Interface (API) (https://portal.gdc.cancer.gov/). Samples without clinical follow-up data or clearly defined statuses were eliminated. The Ensembl IDs were transformed to gene symbols, while the average expression of genes with multiple gene symbols was calculated. After screening, a total of 370 primary tumor samples and 50 adjacent non-tumor samples were successfully acquired. The International Cancer Genome Consortium Liver Cancer Japan Project (ICGC-LIRI-JP) dataset, a bulk dataset obtained from A Database of Hepatocellular Carcinoma Expression Atlas (HCCDB, http://lifeome.net/database/hccdb/), included 212 liver cancer samples and used as a validation set.

GSE166635 containing the scRNA-seq data of two HCC tumor samples (sample numbers: GSM5076749 and GSM5076750) was downloaded from the Gene Expression Omnibus (GEO) database. A total of 59 PMRGs were collected from the "REACTOME_METABOLISM_OF_POLYAMINES" pathway in the MSigDB database (http://www.gsea-msigdb.org/).

### The scRNA-seq data preprocessing

The scRNA-seq dataset GSE166635 was preprocessed under rigorous screening criteria. Specifically, each gene should be expressed in three cells at least, and each individual cell expressed a minimum of 200 genes. Cells were retained if they had 200 to 8,000 genes (nFeature_RNA), <100,000 transcripts (nCount_RNA), and <15% mitochondrial gene expression (percent.mito). Then, the NormalizeData function (*Stuart et al., 2019*) was used to perform log-transformation on the data, and the FindVariableFeatures function (*Stuart et al., 2019*) was applied to accurately filter highly variable genes. The expression values of all genes were normalized by the ScaleData function (*Stuart et al., 2019*). Utilizing the RunPCA function, principal component analysis (PCA) was performed (*Stuart et al., 2019*). Batch effects among various samples were removed applying the harmony package

(*Korsunsky et al., 2019*), selecting the top 20 PCs for dimensionality reduction by UMAP. Then the cells were clustered into subsets by the FindNeighbors and FindClusters (*Stuart et al., 2019*) and finally annotated to specific cell types utilizing the marker genes provided by the CellMarker2.0 database (*Hu et al., 2022*).

### Single-sample GSEA

The enrichment scores of PMRGs in the TCGA-HCC cohort were computed by single-sample GSEA (ssGSEA) (*Barbie et al., 2009*). Specifically, the relative enrichment of this gene set in each sample was estimated by comparing the gene expression data of each sample in the TCGA-HCC cohort to a specific set of PMRGs. The enrichment score calculated by ssGSEA indicated the extent to which a particular gene set within a sample was either synchronously upregulated or downregulated.

### Validation of molecular subtypes classified using the PMRGs

Univariate Cox regression analysis was utilized to elucidate the relationship between PMRGs and the prognosis of HCC. By setting the criterion of $p < 0.05$, PMRGs with significant prognostic significance were screened to classify molecular subtypes of HCC. Subsequently, the ConsensusClusterPlus method (*Wilkerson & Hayes, 2010*) was employed to construct a consensus matrix for accurate molecular subtyping. The clustering process utilized the "km" algorithm and 500 bootstraps, using "1-Spearman correlation" as the distance metric with 80% of the training set patients in each bootstrap. The cluster number was between 2 and 10 for analysis. Prognostic differences among various molecular subtypes were compared by the Kaplan–Meier (KM) curves, and differences in clinicopathological features (age (>60/≤60), stage (I–IV), gender (male/female), T stage (T1–T4), and grade (G1–G4)) of different molecular subtypes in the TCGA-HCC dataset were also explored.

### Discovery of DEGs by enrichment analyses

The DEGs between the identified subtypes were selected by the limma package (*Ritchie et al., 2015*; *Wang et al., 2024*) under the criteria of FDR < 0.05 and |log2FC|>log2(2). Subsequently, using the R package clusterProfiler (*Yu et al., 2012*), KEGG and GO (in biological process, GO-BP) functional enrichment analyses were performed on these DEGs (*Wang et al., 2025*).

### Biomarkers screening through machine learning

Prognostically significant genes ($p < 0.05$) for HCC were selected from the DEGs by univariate Cox regression analysis and then subjected to Lasso regression analysis using the glmnet R package (*Friedman et al., 2021*). Based on the results of Lasso analysis, stepwise multiple regression analysis was further performed. Akaike information criterion (AIC) was utilized to optimize the model during the process. Specifically, the stepAIC method from the MASS package (*Ripley et al., 2013*) was adopted, iteratively removing one variable at a time beginning with the most complex model while monitoring the changes in AIC value.

## Establishment of the risk model

The formula RiskScore = $\Sigma\beta i \times Expi$ was used to compute RiskScore for each patient, with Expi and $\beta$ presenting the gene expression and the Cox regression coefficient of the gene. Subsequently, the RiskScore was normalized by z-score, and high-risk and low-risk groups were divided by the threshold value of "0". The KM survival curve was used to conduct prognostic analysis, and significant differences were analyzed by the log-rank test. Furthermore, the timeROC R package (*Blanche, Dartigues & Jacqmin-Gadda, 2013*) was employed as a prognostic classification tool to conduct a receiver operating characteristic (ROC) analysis for the RiskScore.

## Construction of a nomogram

Univariate and multivariate Cox regression analyses were carried out in the TCGA-HCC dataset to explore whether the RiskScore was independent of Stage, T.stage, N.stage, M.stage, Grade, Risktype, gender. Additionally, survival and risk evaluation for patients in this dataset was further improved by developing a nomogram.

## Relation between the RiskScore and immune microenvironment

Tumor Immune Estimation Resource (TIMER) (*Li et al., 2020*), MCP-counter (*Becht et al., 2016*), and Cell Identification by Estimating Relative Subsets of RNA Transcripts (CIBERSORT) (*Chen et al., 2018*) were employed to examine the status of various immune cell types, so as to clarify the inherent connection between the RiskScore and immune microenvironment of different patients.

## Correlation analysis between drug sensitivity and the RiskScore

The $IC_{50}$ of drugs for the TCGA-HCC dataset samples was predicted by the oncoPredict R package (*Maeser, Gruener & Huang, 2021*). Next, the relationship between the RiskScore and drug sensitivity was examined applying by Spearman's correlation analysis, with a $p$-value < 0.05 and |cor|>0.4 denoting a significant relationship.

## Cell culture

Human liver immortalized cells THLE-2 (C5664), human HCC cells HuH-7 (C5176) were obtained from BDBio (https://www.biocode.cn/) (Hangzhou, China). All the cells used were identified by short tandem repeat (STR) to confirm whether they were contamination-free. Cells were cultured using DMEM (BDBio, Hangzhou, China) with 10% fetal bovine serum (FBS) (C0226; Beyotime, Shanghai, China) and 1% penicillin-streptomycin (P/S) (15140148; Thermo Fisher Scientific, Waltham, MA, USA) and incubated at 37 °C in the concentration of 5% $CO_2$ and saturated humidity.

## RNA extraction and qRT-PCR experiment

Following the instructions, total RNA was separated from THLE-2 and HuH-7 cell samples using Total RNA Extraction Kit (R1200; Solarbio, Beijing, China). RNA quality was assessed by measuring the OD260/OD280 ratio with a NanoDrop microspectrophotometer (Thermo Fisher Scientific, Waltham, MA, USA). Next, the cDNA template was obtained using BeyoRT$^{TM}$ II M-MLV reverse transcriptase (D7160;

Beyotime, Shanghai, China). Changes in the expressions of the biomarkers obtained from computational analysis of THLE-2 and HuH-7 cells were analyzed employing BeyoFast[TM] SYBR Green qPCR Mix (D7265; Beyotime, Shanghai, China) and qRT-PCR technology on an ABI7300 Realtime PCR System (Thermo Fisher Scientific, Waltham, MA, USA). Gene expression was quantified by $2^{-\Delta\Delta CT}$ method and normalized to that of *GAPDH*. The primers were designed by Origene (Rockville, MD, USA). See Table S1 for the relevant sequences and primers.

## Cell transfection

HuH-7 cells were transfected with siRNA by the use of Lipofectamine 2000 Transfection Kit (11668027; Invitrogen, Carlsbad, CA, USA). Following the instructions, Lipofectamine 2000 and siRNA were diluted to 100 nM and 3 μL/well, respectively, to ensure an optimal transfection. Next, the cells were incubated at 37 °C for 48 hours (h). To prevent off-target effects, two siRNA targeting sequences and a negative control (designed and purchased from Merck KGaA, Darmstadt, Germany) with the optimal transfection efficiencies were used for subsequent experiment, which were labeled as si-*G6PD*#1 (target sequence: 5′-GCCTTCCATCAGTCGGATACA-3′), si-*G6PD*#2 (target sequence: 5′-GCTGACAT CCGCAAACAGAGT-3′), and si-NC.

## Cell viability assay

Cell proliferation of HuH-7 cells transfected with si-NC and si-*G6PD* was detected using CCK-8 assay kit (C0037; Beyotime, Shanghai, China). The cells were inoculated at a density of $5 \times 10^3$ cells/well and 100 μL of medium was added to each well to ensure cell growth in a suitable environment. During cell culture, CCK-8 solution (10 μL) was supplemented to each 96-well plate at different time points (24 h, 48 h, 72 h) and incubated for 30 minutes (min) to react with the cells. Finally, the absorbance was measured at 450 nm.

## Cell migration assay

HuH-7 cells were evenly seeded into 6-well plates, with 2 mL of complete medium in each well. The cells were incubated at 37 °C with 5% $CO_2$ until a confluent monolayer was formed. When the cells reached approximately 90% confluence, a straight scratch was gently created across the cell monolayer using a sterilized 200 μL pipette tip. The existing medium was then removed, and the wells were washed twice with PBS (C0221A; Beyotime, Shanghai, China) to eliminate cell debris. Then fresh serum-free medium was added, and the plates were returned to the incubator under the original conditions to continue the incubation. The plates were removed at 0 and 48 h. Photographs of the scratched area were taken with a microscope (Olympus Corporation, Tokyo, Japan), and the time and abnormality were recorded. Finally, changes in scratch width were measured by ImageJ software and the scratch closure (%) was calculated at each time point by the formula: initial scratch width-current scratch width/initial scratch width × 100%.

## Cell invasion capacity

Cell suspension was seeded into the upper Transwell chamber (8 μm, Corning, Corning, NY, USA) at a density of $3 \times 10^4$ cells/chamber, while complete medium was added to the lower chamber as a chemoattractant. After 48-h incubation at 37 °C with 5% $CO_2$, non-invaded cells were removed. The membrane was then washed with PBS (C0221A; Beyotime, Shanghai, China), immobilized in 4% paraformaldehyde (P0099; Beyotime, Shanghai, China) for 15 min, and subsequently colored by crystal violet solution (C0121; Beyotime, Shanghai, China) for 10 min. After rinsing in distilled water, the membrane was air-dried and the invaded cells were quantified from at least five random fields using an inverted microscope (Olympus Corporation, Tokyo, Japan).

## Statistical analysis

R software (version 3.6.0) and GraphPad Prism (version 8.0.2) were used for statistical analyses. Student's t-test or Wilcoxon rank-sum test was used to compare differences in continuous variables between two groups. For datasets involving three or more variables, ANOVA was employed to assess the overall differences among the groups. Subsequently, pairwise comparisons were performed using Sidak's multiple comparison test. Statistically significant was defined $p < 0.05$, ns meant not significant.

# RESULTS

## Molecular subtyping using prognostically significant PMRGs

In the TCGA-HCC cohort, ssGSEA calculation showed that the enrichment scores of PMRGs were higher in the cancer tissue scores than in normal tissues (Fig. 1A, $p < 0.05$). A total of 42 PMRGs ($p < 0.05$) were found to be closely associated with the prognostic outcomes of HCC, and most of them were risk genes. Next, based on the expression profiles of these 42 PMRGs, the consensus clustering analysis revealed a comparatively stable clustering result at $k = 2$, which classified two molecular subtypes (C1 and C2) (Fig. 1B). Notably, the prognosis of C2 was more unfavorable than C1 (Fig. 1C, $p < 0.05$). Meanwhile, the distribution of various clinical features across different molecular subtypes in the TCGA-HCC dataset was compared. It was found that subtype C1 had a relatively more patient at higher clinical grades (Fig. 1D).

## Functional analysis between the two molecular subtypes

We screened 100 DEGs between the subtypes C1 and C2 (Fig. 2A) and analyzed the distribution of these DEGs between the two subtypes (Fig. 2B). KEGG analysis demonstrated that the DEGs were mainly enriched in pathways, including primary bile acid (BA) biosynthesis, PPAR signaling pathway, bile secretion, IL-17 signaling pathway, phenylalanine metabolism (Fig. 2C). GO-BP enrichment analysis showed that these genes were mainly enriched in pathways, including steroid catabolic process, steroid metabolic process, steroid biosynthetic process, cellular hormone metabolic process, BA biosynthetic process (Fig. 2D).

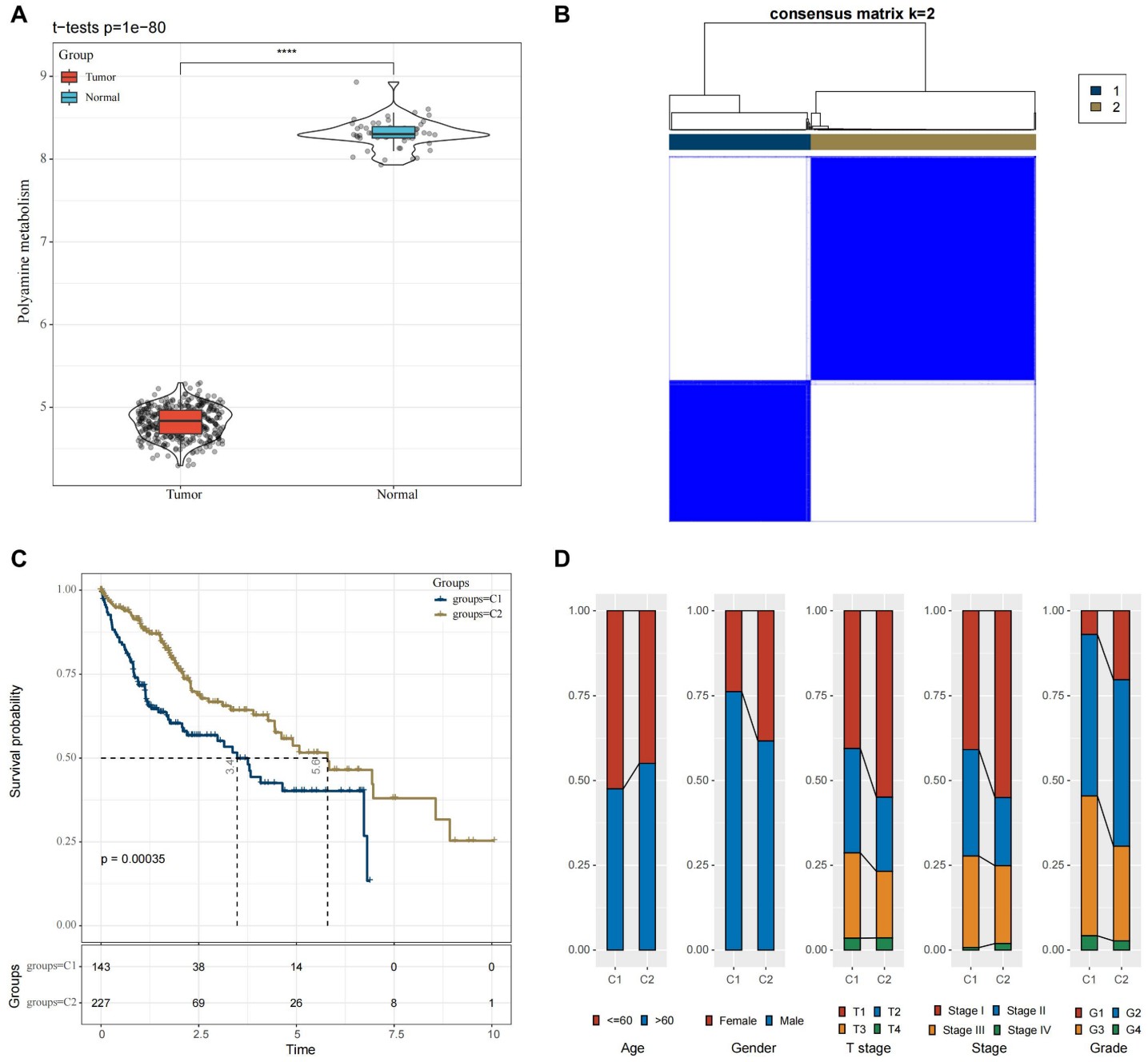

**Figure 1 Molecular subtyping in the TCGA-HCC dataset.** (A) Box plot of ssGSEA results; (B) Consensus clustering heatmap of this dataset; (C) KM curves reflecting the connection between OS prognosis of C1 and C2; (D) Clinicopathological features of C1 and C2. **** means $p < 0.0001$.

## Development of a clinical prognostic model and verification

Univariate COX regression analysis identified 70 genes that may markedly influence HCC prognosis ($p < 0.05$). To optimize the model, Lasso regression demonstrated that variable coefficients gradually approached zero with increasing λ values (Fig. 3A). The optimal performance of the model was achieved at lambda = 0.0674 with 10-fold cross-validation,
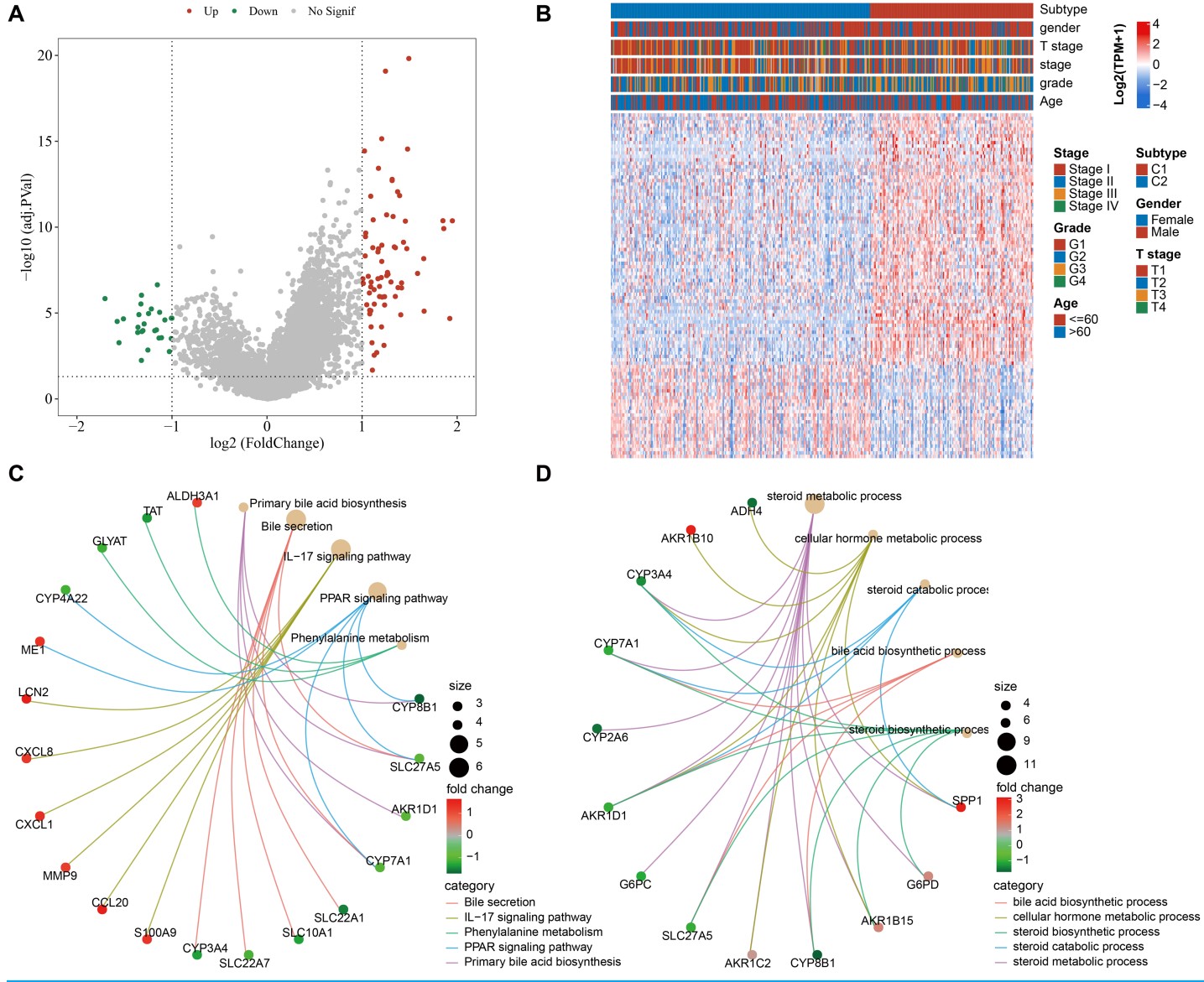

**Figure 2 Functional analysis between the two molecular subtypes.** (A) DEGs in samples of C1 and C2 molecular subtypes were visualized into volcano plot; (B) THLE-2: Distribution of DEGs between C1 and C2; (C) Network diagram between the top 5 pathways and DEGs (KEGG); (D) The diagram of the network between the top 5 GO terms and DEGs (GO-BP).

identifying six key genes (Fig. 3B). Further stepwise multiple regression analysis reduced the six genes to four genes (Fig. 3C), namely, *G6PD*, *S100A9*, *AKR1B15*, and *ADH4*. A risk model was established as: RiskScore = $0.188*G6PD + 0.093*AKR1B15 + 0.079*S100A9 − 0.059*ADH4$.

Subsequently, each sample in the training set was assigned with a RiskScore, normalized by z-score and classified into high- and low-risk groups, which contained 157 and 213 samples, respectively (Fig. 3D). The ROC curve showed a high AUC value of the RiskScore, indicating its robust predictive performance (Fig. 3E). Next, comparison on the overall survival (OS) rates of the patients with different RiskScores demonstrated a negative

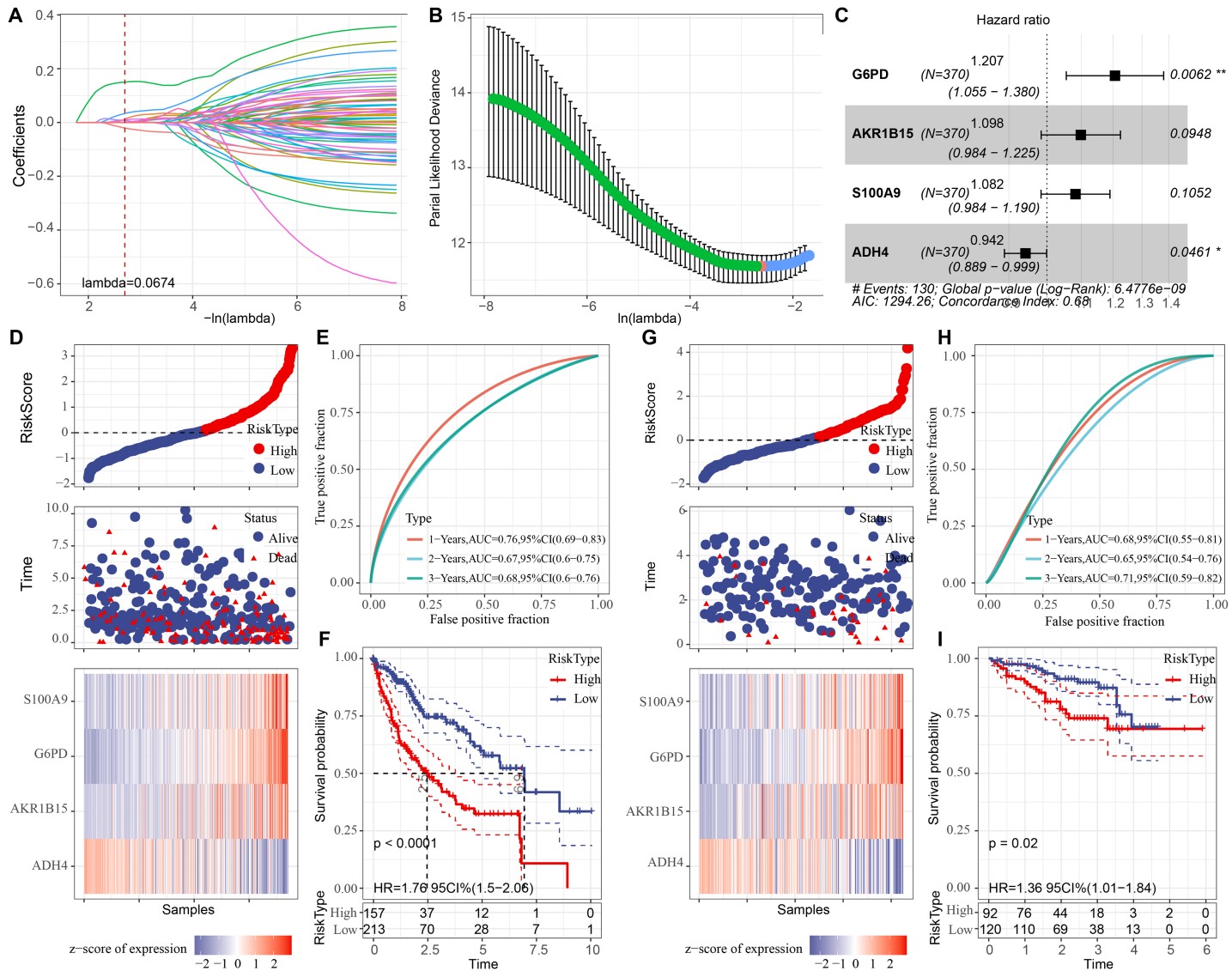

**Figure 3 Development of the prognostic model and verification.** (A) The trajectory of each independent variable as the lambda changes; (B) Confidence interval under different lambda values; (C) Multivariate forest plot for the genes in the model; (D) Expression of RiskScore, survival time and status, and signatures in the TCGA-HCC training set; (E) Clinical prognostic model distribution in the TCGA-HCC training set; (F) KM survival curve in the TCGA-HCC training set. (G) In the ICGC-LIRI-JP validation set, the expression of RiskScore, survival status and time, and signatures; (H) In the ICGC-LIRI-JP validation set, validation of the clinical prognostic model; (I) KM survival curve based on the ICGC-LIRI-JP dataset. * means $p < 0.05$ and ** means $p < 0.01$.

correlation between a worse survival and a high RiskScore (Fig. 3F, $p < 0.05$). In addition, verification in the validation set cohort showed similar results to those obtained in the training set (Figs. 3G, 3H, and 3I).

## Results of *in vitro* experiments

Cellular validation assays demonstrated that the expressions of *G6PD*, *AKR1B15* and *S100A9* were all notably elevated in HuH-7 cells ($p < 0.05$, Fig. 4A), while the level of *ADH4* was not significantly different in HuH-7 cells and THLE-2 cells ($p > 0.05$, Fig. 4A).

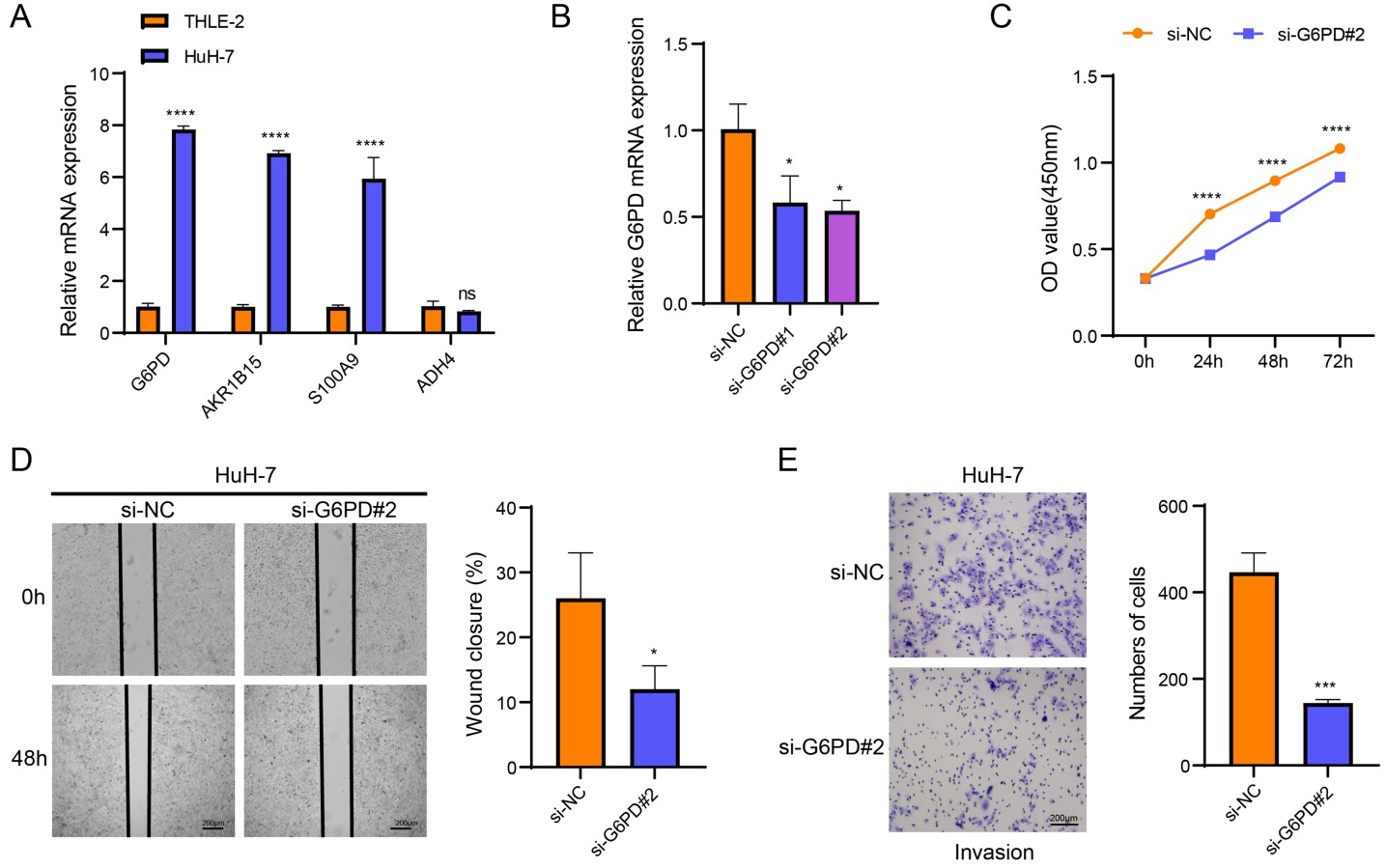

**Figure 4** **Based on *in vitro* cellular assays to assess the expression and potential biological functions of the key genes screened.** (A) Expression of biomarkers in THLE-2 and HuH-7; (B) Validation of transfection efficiency of *G6PD*; (C) CCK8 assay of si-NC and si-*G6PD*#2 group; (D) Wound healing for si-NC and si-*G6PD*#2 group; (E) Transwell assay of si-NC and si-*G6PD*#2 group. ns stands for no significant difference, **** indicates $p < 0.0001$, *** indicates $p < 0.001$, * indicates $p < 0.05$.

Among the four biomarkers, *G6PD* showed the most significant differential expression, and the transfection efficiency of si-*G6PD*#2 was higher than si-*G6PD*#1. Thus, si-*G6PD*#2 was selected for subsequent experiments ($p < 0.05$, Fig. 4B). CCK8 experiments revealed that the cell viability was notably higher in the si-NC group than in the si-*G6PD*#2 group at 24 h, 48 h and 72 h ($p < 0.05$, Fig. 4C). Wound healing and Transwell assays also indicated that the cell migratory and invasive abilities of the si-NC group were higher than those of the si-*G6PD*#2 group ($p < 0.05$, Figs. 4D and 4E). These findings suggested that the silencing of *G6PD* gene expression suppressed the viability, migratory and invasive abilities of HCC cells.

## Development of a nomogram

The RiskScore was verified as the most important prognosis factor ($p < 0.05$) in the TCGA-HCC dataset by univariate and multivariate Cox regression analysis (Figs. 5A and 5B). After developing a nomogram, the RiskScore also manifested the strongest influence on the OS prediction in HCC (Fig. 5C). Further, the calibration curve calibration

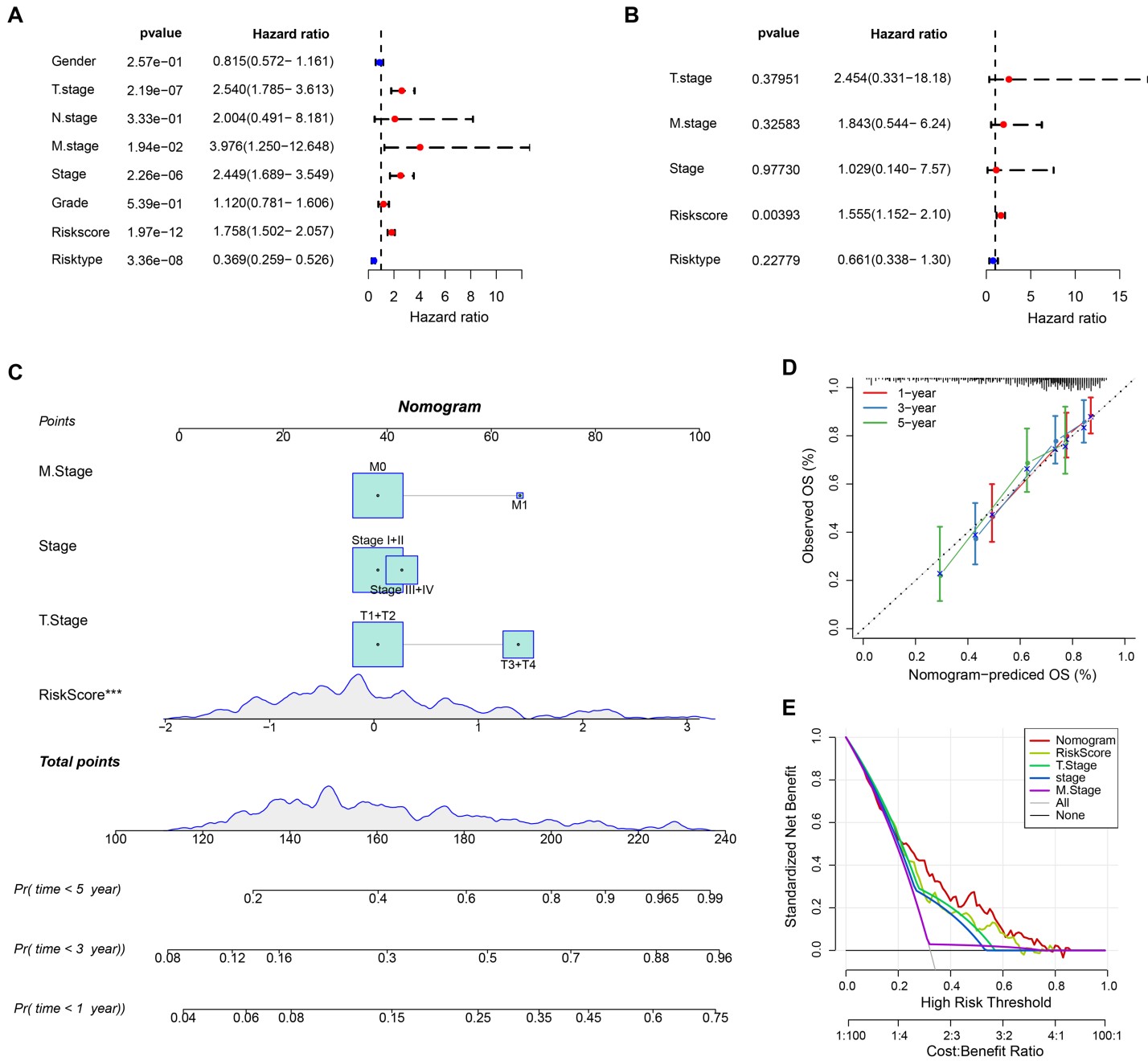

**Figure 5 Nomogram analysis.** (A) Univariate Cox analysis results; (B) Multivariate Cox analysis analysis; (C) Nomogram established by combining RiskScore with clinical features. *** means *p* < 0.001; (D) Calibration curve plotted for the nomogram; (E) DCA of the nomogram.

curves were plotted to test the prediction accuracy of the nomogram, which demonstrated close alignment between the estimated and actual outcomes at 1-, 3-, and 5-year time points (Fig. 5D). This indicated an accurate prediction ability of the nomogram. In addition, the model reliability was assessed by decision curve analysis (DCA). We found

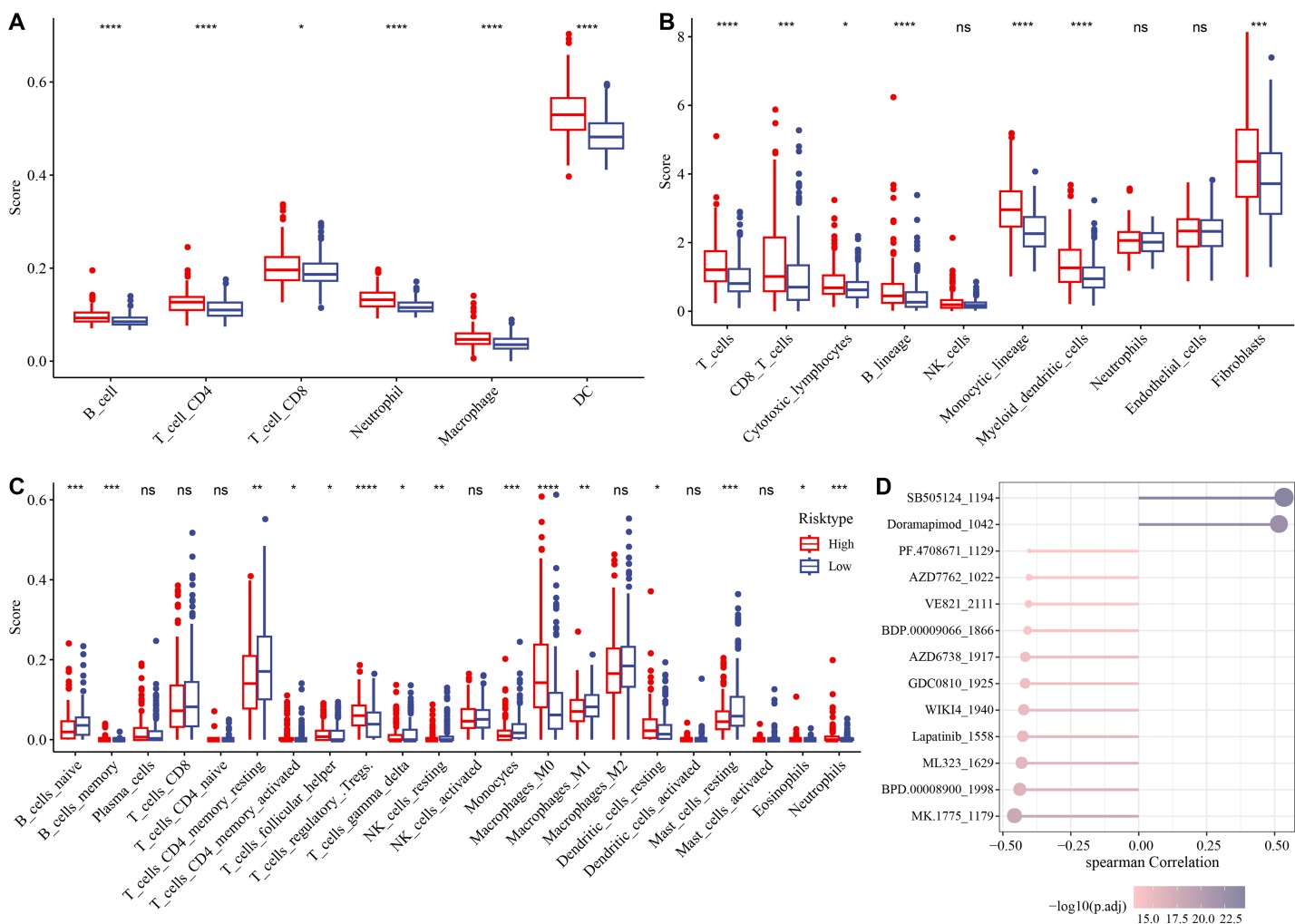

**Figure 6 Analysis of immune microenvironment and drugs ensitivity in the TCGA-HCC cohort.** (A) The differences in immune cell scores between the high-and low-risk groups in the TCGA-HCC cohort in TIMER. * means $p < 0.05$, and **** means $p < 0.0001$; (B) MCP-counter showed immune cell score differences between the two risk groups in the TCGA-HCC cohort. **** means $p < 0.0001$, *** means $p < 0.001$, * means $p < 0.05$; (C) The differences in immune cell scores between the two risk groups in the TCGA-HCC cohort in Cibersort . **** indicates $p < 0.0001$, *** indicates $p < 0.001$, ** indicates $p < 0.01$, * indicates $p < 0.05$, ns means no significant difference; (D) Correlation analysis for RiskScore and drug IC$_{50}$ in the TCGA-HCC dataset.                 

that the nomogram also showed the strongest capability in estimating the OS in HCC than several other clinicopathological features (Fig. 5E).

## Analysis of drug sensitivity and immune microenvironment

The association between immune microenvironment and the RiskScore was comprehensively examined. The results of TIMER indicated that the infiltration of neutrophil cells, dendritic cells (DCs), T cell CD8, macrophages, B cells, T cells CD4 in the high-risk group was significantly higher than those in the control group ($p < 0.05$, Fig. 6A). The MCP-counter further indicated markedly higher scores of most immune cells such as monocytic lineage, T cells, myeloid DCs, CD8 T cells, fibroblasts, B lineages, cytotoxic

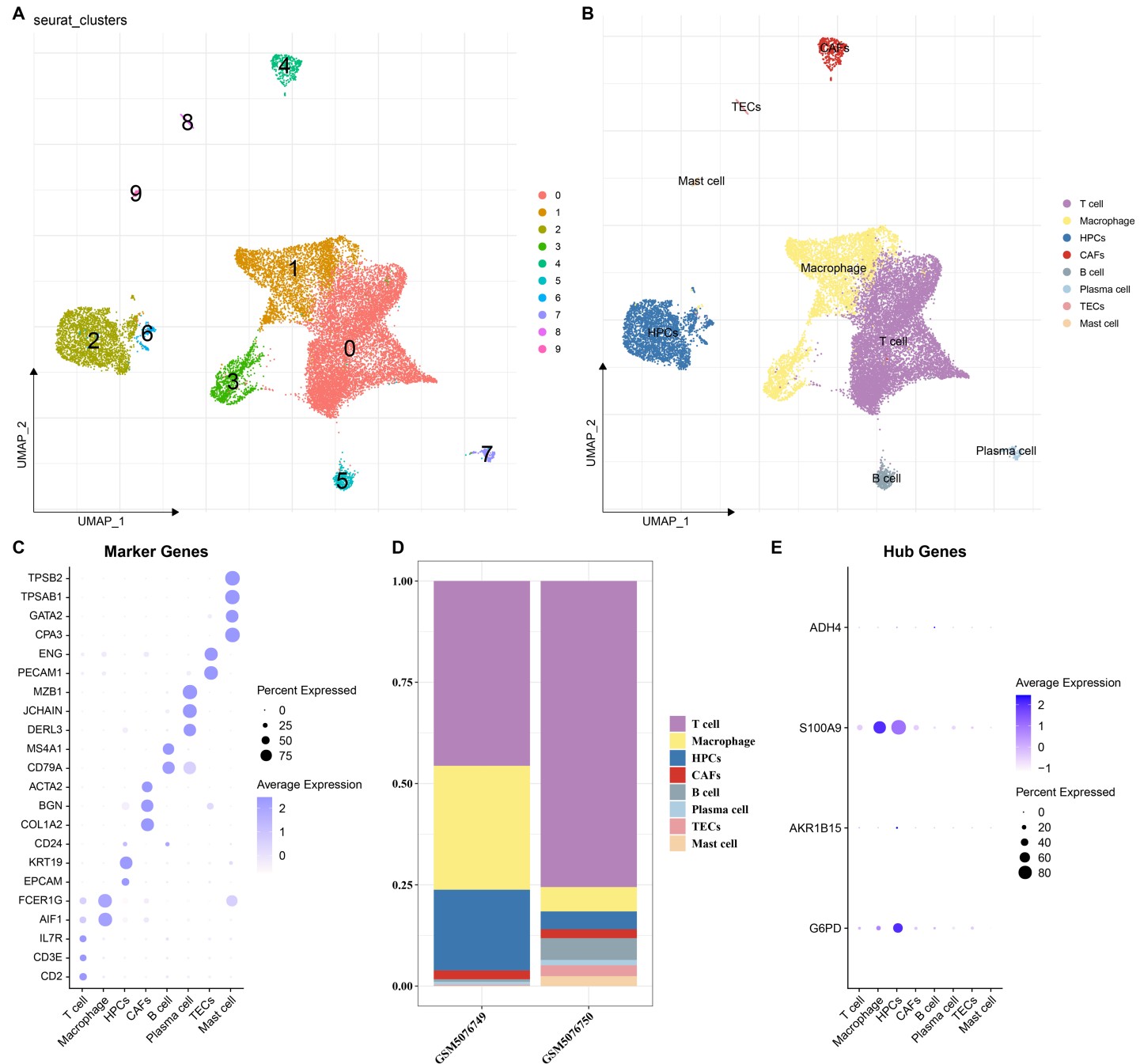

**Figure 7** **Single-cell analysis.** (A and B) UMAP plots of cell subsets before and after annotation; (C) Genes specifically highly expressed in different cell types; (D) Proportions of cell subsets in samples GSM5076749 and GSM5076750; (E) UMAP plots of the expression of *G6PD*, *AKR1B15*, *S100A9* and *ADH4* in different cell clusters.

lymphocytes in the high-risk group than those in the low-risk group ($p < 0.05$, Fig. 6B). CIBERSORT analysis showed that monocytes, B cells naïve, macrophages M1, T cells CD4 memory resting, mast cells resting, T cells gamma delta, *etc.* had markedly higher scores of in the low-risk group, whereas the high-risk group had markedly higher scores of
T cell regulatory Tregs, macrophage M0, T cell follicular helper, DC resting, *etc.* ($p < 0.05$, Fig. 6C).

Subsequently, the IC$_{50}$ values of each drug in the samples of the TCGA-HCC dataset were calculated to identify drugs with significant differences. A significant correlation was found between 13 drugs and the RiskScore. In particular, SB505124_1194 and Doramapimod_1042 were positively correlated with the RiskScore, while the other 11 drugs such as lapatinib_1558 were all negatively correlated with the RiskScore (Fig. 6D).

## Singl-cell atlas

To elucidate the expression distribution of the key genes in different HCC cell types, the scRNA-seq data from two HCC samples (GSM5076749 and GSM5076750) in GSE166635 were analyzed (Figure S1). After cell filtering, normalization, dimensionality reduction, and clustering, a total of 18,369 cells were retained and we clustered 10 major cell subsets. According to the expressions of marker genes in the clusters, eight cell types were annotated, namely T cells, macrophages, cancer-associated fibroblasts (CAFs), B cells, plasma cells, thymic epithelial cells (TECs), mast cells, and HPCs (Figs. 7A and 7B). These 8 cell types were confirmed by identifying their specific marker genes (Fig. 7C). The proportions of cells in the two HCC samples (GSM5076749 and GSM5076750) were different, but both the two were dominated by T cells and macrophages (Fig. 7D). In addition, by analyzing the expressions of the four genes in different cell gene clusters, it was found that both *S100A9* and *G6PD* genes were high-expressed in HPCs and macrophages (Fig. 7E).

## DISCUSSION

HCC is a frequent malignancy and its development is regulated by the immune system (*Donne & Lujambio, 2023*). Polyamines are essential for the proliferation, differentiation, and development of eukaryotes and are closely related to tumorigenesis (*Novita Sari et al., 2021*). Previous studies have found that polyamines and their metabolites could serve as biomarkers for colorectal cancers (*Nakajima et al., 2018*), pancreatic cancer (*Asai et al., 2018*), and kidney cancer (*Li et al., 2023*). In this study, four PMRGs (*G6PD*, *AKR1B15*, *S100A9*, and *ADH4*) were closely related to HCC and validated using *in vitro* experiments. A prognostic model and a nomogram were established based on these biomarkers. Immune infiltration analysis showed that the RiskScore and the immune microenvironment were closely correlated, and we also identified 13 significantly related drugs, providing novel insights into the understanding of HCC and its treatment.

In-depth analysis of the DEGs between the two subtypes revealed a major enrichment of these genes in physiological processes related to BAs. The liver comes into contact with the components of intestinal bacteria and their metabolites through the portal vein (*Yoshimoto et al., 2013*). BAs, amphiphilic molecules produced by the liver, are the key regulators of the gut microbiota and can also be synthesized in the hepatocytes of the liver (*Wang et al., 2013*; *Farhat et al., 2022*). Existing research has demonstrated the BAs both promote and induce the occurrence of HCC, and that abnormally high concentrations of BAs even cause

DNA damage in hepatocytes (*Jia & Jeon, 2019*). At the same time, the gut microbiota uses BAs as signaling molecules to regulate the chemokine-dependent aggregation of natural killer T cells (NK T cells) in the liver, thereby enhancing the anti-tumor immune function of the liver to combat primary and metastatic liver cancers (*Ma et al., 2018*). Based on the above mentioned research findings, the incidence of HCC might be closely linked to BA-related physiological processes.

This study identified four biomarkers, namely *G6PD*, *AKR1B15*, *S100A9*, and *ADH4*. As a crucial rate-limiting enzyme in the pentose phosphate pathway (PPP) (*Meng et al., 2022*; *Ma et al., 2021*), *G6PD* maintains intracellular redox homeostasis and promotes tumor growth (*Cai et al., 2015*). It has been found that *G6PD* can regulate glucose metabolism, affect the development of HCC (*Barajas et al., 2018*), and that its high expression as a marker of poor prognosis is closely associated with HCC metastasis (*Li et al., 2024*). In addition, putrescine in polyamines reduces *G6PD* activity (*ÇİˈFtÇİˈ et al., 2003*). *AKR1B15* is a newly discovered retinol reductase with high activity and shares 91% identity with *AKR1B10* (*Weber et al., 2015*). *AKR1B10* is known to be a biomarker for HCC (*DiStefano & Davis, 2019*), and its high expression is related to increased polyamine pathway activity. S100A9, a protein with multiple ligands and capable of post-translational modification, is involved in both inflammatory responses and cancer cell metastasis (*Chen et al., 2023*; *Markowitz & Carson, 2013*). Inhibition of *S100A9* expression suppresses the progression of HCC through the regulation of M2 macrophage polarization of tumor-associated macrophages (TAMs) (*Yang, Zhang & Wang, 2023*). *ADH4*, a crucial members of the alcohol dehydrogenase family, can metabolize various substrates such as ethanol and retinol and is critical for liver metabolism (*Wei et al., 2012*). *ADH4* shows a notably lower expression in HCC tissue than in normal liver tissue (*Liu et al., 2020*). Given the roles of *ADH4* and *AKR1B15* in the metabolism of retinol-related substances (*Giménez-Dejoz et al., 2015*) as well as the association between reduced serum retinol and early HCV (*Lai et al., 2014*; *Kataria et al., 2016*), these two genes might jointly affect retinol metabolism to influence the development of liver diseases. In addition, the single-cell analysis showed that *S100A9* and *ADH4* were high-expressed in macrophages. Previous studies reported a significant negative association between *ADH4* expression and M1 macrophages (*Li et al., 2024*). This suggested that *S100A9* and *ADH4* genes may interact synergistically to affect macrophages, thereby influencing the development of HCC. In summary, these biomarkers were closely related to polyamine metabolism and affect the occurrence of HCC.

This study found a higher abundance of most immune cells in the high-risk group than in the low-risk group, suggesting multiple potential implications. The higher B-cell score in the high-risk group revealed a more complex immune condition against HCC in this group. On one hand, B cells can actively regulate immune responses and inflammatory reactions by producing antibodies, identify and eliminate antigens such as tumor cells, thus playing a positive role in immunity (*Sarvaria, Madrigal & Saudemont, 2017*). On the other hand, previous studies on immune cells in breast cancer mice showed that regulatory B cells (Bregs) can induce the conversion of T cells into regulatory T cells (Tregs), promoting the formation of an immunosuppressive state in the TME (*Olkhanud et al., 2011*). In the

current research, an increase in the Tregs score in the high-risk group suggested an inhibited immune response , allowing tumor cells to escape immune surveillance of the body and continue to grow and spread (*Qiu et al., 2022*; *Chen et al., 2022*). In addition, increase in cells such as neutrophils, macrophages, *etc.* in the high-risk group indicated a strong inflammatory microenvironment in this group. During cancer development, macrophages are reprogrammed into TAMs, presenting different functional polarization states such as M1 (anti-inflammatory) and M2 (pro-inflammatory) phenotypes (*Kohlhepp et al., 2023*). The findings of this research showed that the low-risk group had a higher M1 macrophage score, which could potentially explain its more favorable prognosis. In summary, the distinct immune microenvironment between the two risk groups indicated that patients in these two groups should be clinically treated differently.

In addition, we identified 13 drugs associated with the RiskScore. Doramapimod was positively correlated with the RiskScore, while Lapatinib was negatively correlated with the RiskScore. Lapatinib, an oral dual tyrosine kinase inhibitor, can effectively prevent the M2 polarization of macrophages. Previous discovery reveals a potential mechanism for inhibiting lung cancer growth through interventions targeting M2 polarization (*Tariq et al., 2023*). Doramapimod is a type-II inhibitor that demonstrates both high potency and exceptional selectivity for human p38α MAPK through inhibiting the p38 MAPK signaling pathway to suppress the production of inflammatory mediators (*Moon et al., 2019*). Again, these results suggested that different agent may be used for treating the two groups.

Some limitations in the current work should be noted. Firstly, the database size was comparatively small and may not fully represent the genetic and phenotypic diversity of HCC patients. Secondly, *in vitro* experiments revealed a downregulation trend of *ADH4* expression in HCC but there was no significant difference, which requires further *in vivo* experimental validation. Additionally, the safety and efficacy of the predicted drugs should be tested following the standardized clinical trial procedures. To address these issues, animal models are encouraged to evaluate the effects of the biomarkers, the anti-tumor efficacy, toxicity, and the impacts of the predicted drugs on the immune microenvironment. Meanwhile, the database analyzed should be expanded by collecting more multi-dimensional data from HCC patients to improve the reliability of the research.

## CONCLUSIONS

This research identified PMRGs closely related to HCC, namely *G6PD*, *AKR1B15*, *S100A9*, and *ADH4*, and established a prognostic model along with a nomogram. This research provided novel biomarkers and potential therapeutic drugs for HCC. The prognostic model and nomogram can assist the prediction of the HCC progression and guide clinical decision-making. The novel insights on the immune microenvironment could enhance our understanding of the immune regulation mechanism in HCC and facilitate the improvement of immunotherapies. Overall, this study provided new potential targets for HCC treatment, facilitating stratified treatment guided by risk stratification for HCC patients.

## ABBREVIATIONS

| | |
|---|---|
| HCC | hepatocellular carcinoma |
| HCV | hepatitis C virus |
| eIF5AH | hypusinated eukaryotic translation initiation factor 5A |
| TCA | tricarboxylic acid |
| OXPHOS | oxidative phosphorylation |
| PMRG | polyamine metabolism related gene |
| TCGA | The Cancer Genome Atlas |
| GDC | Genomic Data Commons |
| RNA-Seq | RNA sequencing |
| API | Application Programming Interface |
| HCCDB | A Database of Hepatocellular Carcinoma Expression Atlas |
| ICGC-LIRI-JP | International Cancer Genome Consortium Liver Cancer Japan Project |
| GEO | Gene Expression Omnibus |
| scRNA-seq | single-cell RNA sequencing |
| PCA | principal component analysis |
| UMAP | uniform manifold approximation and projection |
| ssGSEA | single-sample gene set enrichment analysis |
| KM | Kaplan–Meier |
| FDR | false discovery rate |
| DEG | differentially expressed gene |
| KEGG | Kyoto Encyclopedia of Genes and Genomes |
| GO | Gene Ontology |
| BP | biological process |
| AIC | Akaike Information Criterion |
| ROC | receiver operating characteristic |
| IC50 | half maximal inhibitory concentration |
| STR | short tandem repeat |
| DMEM | Dulbecco's Modified Eagle Medium |
| FBS | fetal bovine serum |
| P/S | penicillin-streptomycin |
| qRT-PCR | quantitative reverse transcription polymerase chain reaction |
| ANOVA | analysis of variance |
| ns | not significant |
| BA | bile acid |
| OS | overall survival |
| DCA | decision curve analysis |
| DC | dendritic cell |
| HPC | hematopoietic progenitor cell |
| CAF | cancer-associated fibroblast |

| TEC | thymic epithelial cell |
|---|---|
| **NK T cell** | natural killer T cell |
| **G6PD** | glucose-6-phosphate dehydrogenase |
| **PPP** | pentose phosphate pathway |
| **AKR1B15** | aldo-keto reductase family 1 member B15 |
| **AKR1B10** | aldo-keto reductase family 1 member B10 |
| **S100A9** | S100 calcium binding protein A9 |
| **TAM** | tumor-associated macrophage |
| **ADH4** | alcohol dehydrogenase 4 |
| **Breg** | regulatory B cell |
| **Treg** | regulatory T cell |
| **MARK** | mitogen-activated protein kinase |

### Funding

This study is supported by Clinical Research Project of Air Force Medical Center (2021LC013) and Youth Science and Technology Support Project of Air Force Medical Center (22YXQN042). The funders had no role in study design, data collection and analysis, decision to publish, or preparation of the manuscript.

### Grant Disclosures

The following grant information was disclosed by the authors:
Air Force Medical Center: 2021LC013 and 22YXQN042.

### Competing Interests

The authors declare that they have no competing interests.

### Author Contributions

- Chengli Liu conceived and designed the experiments, analyzed the data, prepared figures and/or tables, authored or reviewed drafts of the article, and approved the final draft.
- Meng Pu conceived and designed the experiments, performed the experiments, prepared figures and/or tables, authored or reviewed drafts of the article, and approved the final draft.
- Yingbo Ma performed the experiments, prepared figures and/or tables, and approved the final draft.
- Shuhan Zhang conceived and designed the experiments, prepared figures and/or tables, and approved the final draft.
- Qike Huang performed the experiments, prepared figures and/or tables, and approved the final draft.
- Haoming Li performed the experiments, analyzed the data, authored or reviewed drafts of the article, and approved the final draft.

- Tian Xia analyzed the data, authored or reviewed drafts of the article, and approved the final draft.
- Jingchen Zhang analyzed the data, authored or reviewed drafts of the article, and approved the final draft.

## Data Availability

The datasets are available at GEO: GSE166635.

The raw data is available in GitHub and Zenodo:

- https://github.com/liuchengli370/data.git

- liuchengli370. (2025). liuchengli370/data: Raw data (v1.1.0). Zenodo. https://doi.org/10.5281/zenodo.15240263.

## Supplemental Information

Supplemental information for this article can be found online at http://dx.doi.org/10.7717/peerj.19985#supplemental-information.

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
