# Peer review of "Predicting the prognosis of hepatocellular carcinoma based on genes related to polyamine metabolism"

_PeerJ, doi:10.7717/peerj.19985_

## Round 0.1 · original submission · Major Revisions

After considering all reviewers’ comments, I have decided that your manuscript requires major revision before it can be reconsidered for publication. Please address all reviewer comments carefully.

**Language Note:** The review process has identified that the English language must be improved. PeerJ can provide language editing services - please contact us at [email protected] for pricing (be sure to provide your manuscript number and title). Alternatively, you should make your own arrangements to improve the language quality and provide details in your response letter. – PeerJ Staff

Reviewer 1 ·

Basic reporting

This study integrated polyamine metabolism and prognosis modeling of liver cancer, and combined with single-cell data to analyze cellular heterogeneity, which is somewhat innovative. In addition, the risk model and Nomogram have potential clinical application prospects, but the following problems need to be solved.
1. The methodology in the abstract is too lengthy. Please summarize and simplify it to reduce the word count.
2. The content of the Conclusion in the abstract seems more like the aim of this paper. Please add what the conclusion of this study is.
3. The Background in the abstract also needs to reflect the purpose of this paper. Moreover, the background needs to be supplemented to be more engaging, so that readers can clearly understand the significance of this study.
4. Please specify the exact web address of the database.
5. This paper needs to be polished to be more fluent and professional.
6. The references are outdated. Please try to cite literature from the past three years.
7. Each figure has two captions, and the captions are not consistent. For example, the caption above Figure 4 does not indicate the meaning of the significance symbols.
8. The meaning of "ns" in Fig6c still needs to be added.
9. This study only experimentally verified G6PD. Since this research focuses on verifying and experimenting on the function of G6PD, both the abstract and the title of this paper need to be revised.
10. Why was only G6PD selected for the experiment? I suggest verifying other genes as well.
11. Some sentences have grammatical errors or unclear expressions (for example, "the scores of most immune cells were higher in the high-risk group" needs to specify the specific cell type). Language polishing is recommended.

Experimental design

None

Validity of the findings

None

Reviewer 2 ·

Basic reporting

no comment

Experimental design

no comment

Validity of the findings

no comment

Additional comments

This study systematically analyzed polyamine metabolism-related genes in hepatocellular carcinoma (HCC), identified molecular subtypes associated with prognosis, screened key biomarkers, and constructed a risk model. It revealed their roles in tumor progression and the immune microenvironment, providing a theoretical basis for precision therapy and drug development in HCC.
1. Line 23-24, In the background, only HCC was introduced, lacking a description of the purpose of the article. Line 25-26, which datasets are used in this study. In methods of abstract, Stepwise multiple regression analysis is a standard statistical term. It is suggested to sort out the logical sequence of the methods in abstract section. “Cibersort” should be capital.
2. The abstract's Methods section is overly detailed and repetitive, lacking a clear structure. It should focus on key steps like subtype identification, biomarker screening, and model building, using a more concise and integrated summary of the analytical approach. In results, the biomarkers should be simplified for the more important content, increasing the information density of the abstract.
3. Line 59-60, Antiviral therapy has no inevitable connection with the increased incidence of diseases. Line 60-63, Why identifying relevant biomarkers is significant importance, there is a lack of necessary transitions.
4. Line 64-66, it is suggested sentence merging. What are the characteristics of polyamine metabolism. What are the molecular mechanisms and pathway signals that affect abnormal polyamine metabolism. Line 67-71, This paragraph describes how polyamine metabolism promotes M2 macrophage polarization via eIF5AH and mitochondrial pathways, but its relevance to HCC is unclear. A clearer link between M2-like tumor-associated macrophages and HCC progression is needed to strengthen the connection and maintain focus.
5. Line 76-79, Considering the crucial role of polyamine metabolism in HCC, the mechanisms by which polyamine metabolism affects the progression of HCC are too few to highlight this significance. Line 79-88, It is suggested to improve logic.
6. Line 91, bulk dataset and the single-cell dataset are described separately. Little title of “2.1” and “2.2 Data preprocessing”, the title content is repetitive. Line 94-95, “specimens” and “samples” should be unified. Line 105-108, The screening method of RNA-seq and the dataset are placed together. The content has poor logic and serious redundancy. It is suggested to adjust in the Material and methods section. Line 229, for “pipette gun”, Don't translate directly. Line 280-281, result descriptions are generally in the passive voice. Line 295, Please explain the rationality of conducting z-score.
7. Line 345-346, The purpose of single-cell analysis is not clear. Please adjust it.
8. Line 379, the HCC is closely linked to BA-related physiological processes. What is the connection between bile salts and polyamine metabolism.
9. What is the specific clinical application of RiskScore and how does it improve patient outcomes.
10. This article has extremely poor logic and the sentence quality is poor, the whole text needs to be polished. And the resolution of the picture needs to be improved.

---

## Round 0.2 · accepted · Accept

Thank you for submitting your revised manuscript and for addressing the reviewers' comments thoroughly. Both reviewers have recommended acceptance, and I am pleased to inform you that your manuscript has been accepted for publication.

Reviewer 1 ·

Basic reporting

This study integrated polyamine metabolism and prognosis modeling of liver cancer, and combined with single-cell data to analyze cellular heterogeneity, which is somewhat innovative. In addition, the risk model and Nomogram have potential clinical application prospects. Despite the limitation of small sample size, the research design was rigorous and the data was transparent, providing important theoretical basis for the molecular classification and precise treatment of HCC. I recommend publication.

Experimental design

no comment

Validity of the findings

no comment

Reviewer 2 ·

Basic reporting

This study systematically analyzed polyamine metabolism-related genes in hepatocellular carcinoma (HCC), identified molecular subtypes associated with prognosis, screened key biomarkers, and constructed a risk model. It revealed their roles in tumor progression and the immune microenvironment, providing a theoretical basis for precision therapy and drug development in HCC

Experimental design

This study systematically analyzed polyamine metabolism-related genes in hepatocellular carcinoma (HCC), identified molecular subtypes associated with prognosis, screened key biomarkers, and constructed a risk model. It revealed their roles in tumor progression and the immune microenvironment, providing a theoretical basis for precision therapy and drug development in HCC

Validity of the findings

This study systematically analyzed polyamine metabolism-related genes in hepatocellular carcinoma (HCC), identified molecular subtypes associated with prognosis, screened key biomarkers, and constructed a risk model. It revealed their roles in tumor progression and the immune microenvironment, providing a theoretical basis for precision therapy and drug development in HCC